# Multiple independent genetic code reassignments of the UAG stop codon in phyllopharyngean ciliates

**Jamie McGowan** [1] *, **Thomas A. Richards** [2], **Neil Hall** [1,3], **David Swarbreck** [1] *

**1** Earlham Institute, Norwich Research Park, Norwich, United Kingdom, **2** Department of Biology, University of Oxford, Oxford, United Kingdom, **3** School of Biological Sciences, University of East Anglia, Norwich, United Kingdom

* Jamie.McGowan@earlham.ac.uk (JM); David.Swarbreck@earlham.ac.uk (DS)

**Data Availability Statement:** Supporting data have been deposited on Zenodo (10.5281/zenodo. 12744466).

## Abstract

The translation of nucleotide sequences into amino acid sequences, governed by the genetic code, is one of the most conserved features of molecular biology. The standard genetic code, which uses 61 sense codons to encode one of the 20 standard amino acids and 3 stop codons (UAA, UAG, and UGA) to terminate translation, is used by most extant organisms. The protistan phylum Ciliophora (the 'ciliates') are the most prominent exception to this norm, exhibiting the grfeatest diversity of nuclear genetic code variants and evidence of repeated changes in the code. In this study, we report the discovery of multiple independent genetic code changes within the Phyllopharyngea class of ciliates. By mining publicly available ciliate genome datasets, we discovered that three ciliate species from the TARA Oceans eukaryotic metagenome dataset use the UAG codon to putatively encode leucine. We identified novel suppressor tRNA genes in two of these genomes which are predicted to decode the reassigned UAG codon to leucine. Phylogenomics analysis revealed that these three uncultivated taxa form a monophyletic lineage within the Phyllopharyngea class. Expanding our analysis by reassembling published phyllopharyngean genome datasets led to the discovery that the UAG codon had also been reassigned to putatively code for glutamine in *Hartmannula sinica* and *Trochilia petrani*. Phylogenomics analysis suggests that this occurred via two independent genetic code change events. These data demonstrate that the reassigned UAG codons have widespread usage as sense codons within the phyllopharyngean ciliates. Furthermore, we show that the function of UAA is firmly fixed as the preferred stop codon. These findings shed light on the evolvability of the genetic code in understudied microbial eukaryotes.

## Author summary

The genetic code dictates the translation of nucleotide sequences into amino acids. It is one of the most conserved features of molecular biology. Most organisms use the "standard genetic code", where 61 codons encode amino acids and three stop codons (UAA, UAG, and UGA) signal translation termination. Ciliates–microbial single-celled

**Funding:** This work was funded by the Wellcome Trust Darwin Tree of Life Awards (218328 and 226458 to NH and TAR) and by the Biotechnology and Biological Sciences Research Council (BBSRC), part of UK Research and Innovation, through the Core Capability Grant (BB/CCG2220/1 to NH) at the Earlham Institute; the Earlham Institute Strategic Programme Grant Decoding Biodiversity (BBX011089/1 to NH and DS) and its constituent work packages (BBS/E/ER/230002A and BBS/E/ER/230002B to NH and DS). TAR is supported by a Royal Society University Research Fellowship (URF/R/191005 to TAR). The funders had no role in study design, data collection and analysis, decision to publish, or preparation of the manuscript.

**Competing interests:** The authors have declared that no competing interests exist.

eukaryotes–are the most prominent exception, with multiple lineages exhibiting variant genetic codes in which one or more stop codons have been reassigned to function as sense codons. In this study, we report novel genetic code changes in poorly studied ciliates. By analysing marine metagenomic data from the TARA Oceans Project, we identified an uncultivated lineage of ciliates from the Phyllopharyngea class that uses the UAG codon to encode leucine. Extending our analysis to include other genomes from the Phyllopharyngea class, we identified further changes in *Hartmannula sinica* and *Trochilia petrani* where the UAG codon had been reassigned to encode glutamine. Phylogenomics analysis suggests that three lineages within Phyllopharyngea independently reassigned the codon UAG to encode an amino acid. These findings expand our understanding of genetic code evolution and highlight the remarkable diversity of genetic codes employed by ciliates.

## Introduction

Ciliates are a diverse phylum of single-celled eukaryotes (protists) characterised by their unusual genome biology. Interestingly, ciliate species exhibit the greatest diversity of variant nuclear genetic codes, with several ciliate lineages possessing genetic codes that deviate from the standard genetic code. The standard genetic code uses three stop codons (UAA, UAG, and UGA) to terminate translation and 61 sense codons to encode an amino acid [1]. Once thought to be universal as it is used by most extant organisms, the standard genetic code is one of the most conserved features of molecular biology, emerging prior to the last universal common ancestor (LUCA) [2].

All reported genetic code changes in ciliates involve the reassignment of one or more stop codons to encode an amino acid. The most common variant genetic code in ciliates (and eukaryotes in general) involves the reassignment of both the UAA and UAG (i.e., UAR) codons to encode glutamine, as observed in several lineages of ciliates, including the model ciliate species *Tetrahymena thermophila* and *Oxytricha trifallax* [3]. In comparison, the UAR codons are reassigned to glutamic acid in the peritrichs [4] and to tyrosine in the *Mesodinium* genus [5]. Whereas the UGA stop codon has been reassigned to cysteine in *Euplotes* [6] and to tryptophan in *Blepharisma* and *Condylostentor* [3,7]. In some lineages of ciliates, including Karyorelictea, *Condylostoma* and *Plagiopyla*, all three stop codons have been reassigned and can have dual meanings encoding an amino acid or terminating translation depending on their context [5,8–10]. In genomes with context dependent genetic codes, it has been demonstrated that codons with dual meanings (i.e., stop and sense) are underrepresented in proximity with the true termination codon [9].

In most reported variant genetic codes, the codons UAA and UAG have the same meaning, either they are both reassigned to code for an amino acid or they are both retained as stop codons, suggesting that evolutionary or mechanistic constraints couple the function of these two codons [11,12]. One proposed mechanism for the coupling of UAA and UAG codon meanings is wobble base pairing of suppressor tRNAs. It has been experimentally demonstrated in *Tetrahymena thermophila* that a suppressor tRNA with a UUA anticodon suppresses both UAA and UAG codons due to wobble base pairing [13]. Thus, if the function of the UAA stop codon was reassigned to function as a sense codon via acquisition of a tRNA-Sup(UUA), it would also likely trigger the reassignment of the UAG stop codon to the same amino acid. Conversely, if UAG were reassigned via acquisition of a tRNA-Sup(CUA), it would not be expected to trigger the reassignment of the UAA stop codon, as the CUA anticodon should specifically recognise UAG [13]. Therefore, genetic code variants where UAA and UAG

codons have different meanings are particularly interesting and rare. Such genetic codes require the translational machinery to evolve to efficiently decode one as a sense codon while retaining the other as a termination codon.

In this study, we report the discovery of three independent genetic code change events within the Phyllopharyngea class of ciliates. The Phyllopharyngea class is relatively understudied compared to other ciliate groups and includes taxa with diverse morphologies and lifestyles, including free-living species and symbiotic species [14]. They include some of the most destructive parasites of fish [15]. Mining publicly available ciliate genome sequences, we identified three uncultivated ciliate species from the TARA Oceans eukaryotic metagenome dataset [16] where the UAG stop codon has been reassigned to code for leucine. We identified novel suppressor tRNA genes with CUA anticodons in two of these genomes which are predicted to decode the reassigned UAG codon to leucine. Phylogenomic analysis revealed that these three uncultivated taxa form a monophyletic lineage within the Phyllopharyngea class. Reassembly and annotation of seven other phyllopharyngean genome sequences from previously published datasets [17,18] revealed that *Hartmannula sinica* and *Trochilia petrani* have also undergone genetic code reassignments where UAG has been reassigned to encode glutamine. The other five phyllopharyngean species use the standard genetic code. Phylogenomic analysis infer that the reassignment of the UAG stop codon to encode glutamine has evolved independently twice within the Phyllopharyngea lineage. Thus, Phyllopharyngea contains a mix of species that use the standard and variant genetic codes, with at least three independent genetic code change events. The reassigned UAG codon demonstrates widespread usage in the predicted proteomes as a sense codon in all five species showing reassignment, suggesting that its function is fixed as a sense codon. Furthermore, these data demonstrate that UAA is ubiquitously used as the preferred stop codon in these taxa and is therefore unlikely to later be reassigned as a sense codon. These findings reveal further divergences between the function of the UAA and UAG codons signifying repeat breaking of the proposed mechanistic constraints linking the function of UAA and UAG codons.

## Results

### Reassignment of the UAG stop codon to leucine in an uncultivated lineage of ciliates

To investigate the evolution of the genetic code in uncultivated ciliates, we mined eukaryotic metagenome-assembled genomes (MAGs) from the TARA Oceans project [16]. This is a dataset of manually curated genome assemblies. 30 MAGs from this dataset were classified as being ciliates, which we analysed in this study. Two complementary tools were initially used to predict the genetic code of each genome assembly–Codetta and PhyloFisher [19,20]. Codetta predicts the meaning of each codon by aligning hidden Markov models from the Pfam database against a six-frame translation of a query genome assembly. The "genetic_code_examiner" utility from PhyloFisher predicts the genetic code by identifying and counting codons that correspond to highly conserved amino acid sites in a database of orthologous proteins.

This analysis revealed a novel genetic code change in ciliates. Three of the TARA Oceans ciliate MAGs were predicted to have reassigned the meaning of the UAG codon to encode leucine–TARA_ARC_108_MAG_00274, TARA_ARC_108_MAG_00306, and TARA_SOC_28_-MAG_00066. We will refer to these as ARC_274, ARC_306, and SOC_66, respectively hereafter. Two of these MAGs are from the Arctic Ocean (ARC_274 and ARC_306), and the other MAG is from the Southern Ocean (SOC_66) [16]. ARC_306 (33.6 Mb; 77.8% BUSCO completeness) and SOC_66 (21.3 Mb; 49.8% BUSCO completeness) both have reasonable estimated genome completeness (**S1 Table**). The third MAG ARC_274 has low completeness (11

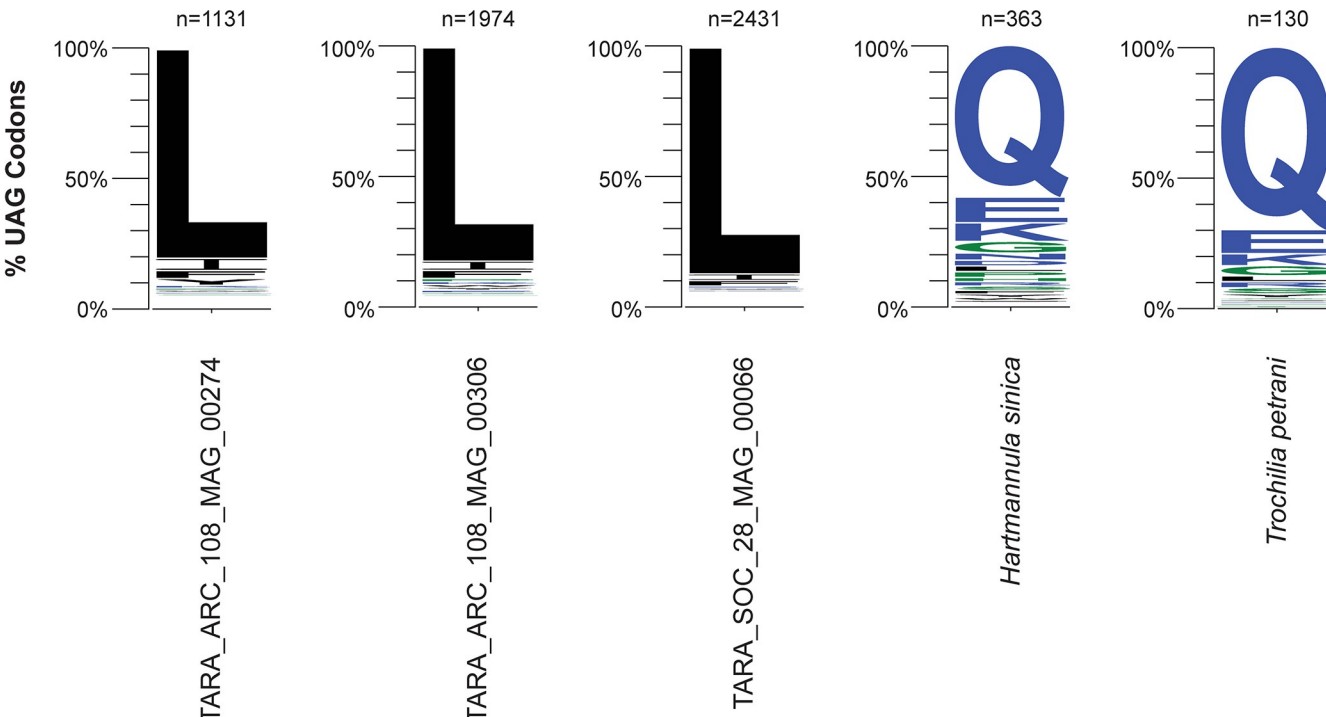

**Fig 1. Genetic code prediction of the UAG codon.** The genetic code of each species was predicted by identifying in-frame UAG codons that occur at highly conserved (≥ 70% identity) amino acid positions in ciliate orthogroups. Each sequence logo represents the frequency of the most numerous amino acids at these highly conserved positions for each species. Numbers represent the number of codons analysed.

Mb; 12.9% BUSCO completeness) which is reflected by its smaller assembly size (**S1 Table**). Codetta identified 1011, 1898, and 2583 UAGs codons with a Pfam alignment (**S2 Table**) in ARC_274, ARC_306, and SOC_66, respectively, and predicted that all three MAGs translate UAG to leucine with a log decoding probability of zero (**S1 Fig**). The number of UAG codons with an alignment was within the range of other sense codons (**S2 Table**). From the Phylo-Fisher dataset of 240 eukaryotic orthologs, PhyloFisher identified 8, 12, and 44 internal UAGs codons in ARC_274, ARC_306, and SOC_66, respectively, which correspond to positions where leucine is highly conserved (> 70% conservation) (**S2 Fig**). This is a very strict analysis as it only considers amino acid positions that are highly conserved across diverse and distantly related lineages of eukaryotes, with no closely related relatives to the taxa from this study. To expand this analysis further, we carried out our own manual analysis employing a similar approach. We annotated each MAG by training a specific Augustus model for each genome (see details below). Using these annotations and a set of ciliate proteomes (**S1 Table**), we generated a set of ciliate orthogroups using OrthoFinder and identified in-frame UAG codons that correspond to highly conserved (≥ 70% identity) amino acid sites and recorded the most numerous amino acid at these sites. This analysis yielded the same predictions as PhyloFisher but with greater support using our ciliate-specific dataset. 1131, 1974, and 2431 in-frame UAG codons that correspond to highly conserved amino acid sites were identified in ARC_274, ARC_306, and SOC_66, respectively, of which 80.9%, 82.8%, and 87.7% corresponded to positions where leucine is highly conserved (**Fig 1**). This genetic code variant (UAG translated to leucine) is represented by translation table 16 and named the "Chlorophycean Mitochondrial Code" in the NCBI database of genetic codes (https://www.ncbi.nlm.nih.gov/Taxonomy/Utils/wprintgc.cgi).

To investigate if UAG potentially has a dual context dependent meaning (i.e., both sense and stop depending on context), we aligned reference proteins from the *Tetrahymena thermophila* proteome against each genome assembly using miniprot and identified alignments where the C-terminus of query proteins aligned to the genome and checked if that position in the genome corresponded to a stop codon. This approach is independent of our genome annotation. When translation table 1 was used (i.e., treating UAA, UAG, and UGA as stop codons), there was a clear difference between taxa that use the standard genetic code and the TARA MAGs with reassigned codes (S3 Table). In the TARA MAGs, 92.6% - 96% of alignments corresponded to UAA stop codons in the genome. Only 1–4 alignments corresponded to UAG stop codons, which was comparable to the number that correspond to other sense codons (S3 Table). When translation table 16 was used for miniprot alignment (i.e., treating UAA and UGA as stop codons and UAG as leucine), the number of alignments where the C-terminus of query proteins aligned increased. This suggests that it is unlikely that UAG is also used as a stop codon. Furthermore, we investigated if the frequency of in-frame UAG codons depleted in proximity with the annotated stop codon in genes and found that this was not the case (S3 Fig). The distribution of UAG codons across the gene body showed a frequency pattern similar to that of synonymous leucine codons (S3 Fig). Thus, there is no evidence to suggest that UAG has a dual context dependent meaning.

An example multiple sequence alignment of the DRG2 protein (developmentally-regulated GTP-binding protein 2) is shown in Fig 2 with ciliate sequences aligned against orthologs from diverse eukaryotic species. The ARC_274 sequence contains three internal UAG codons which correspond to positions where leucine is highly conserved in the other eukaryotic sequences (Fig 2). The ARC_306 sequence contains a single internal UAG codon corresponding to leucine and the SOC_66 sequence contains three internal UAG codons corresponding to leucine (and one to isoleucine) (Fig 2).

A challenge with metagenome binning is that ribosomal rRNA genes are typically missing from MAGs, due to technical limitations, which is the case here making detailed taxonomic identification difficult. Instead, we relied upon the alpha-tubulin protein as a phylogenetic marker. An alpha-tubulin protein was recovered from just one of the MAGs (ARC_306). Phylogenetic analysis placed the ARC_306 sequence within a clade of sequences from the Phyllopharyngea class (S4 Fig) with high support (91% ultrafast bootstrap support). We extend this phylogenetic analysis below using phylogenomic approaches.

## Reassignment of the UAG stop codon to glutamine in *Hartmannula sinica* and *Trochilia petrani*

To expand our dataset further, we retrieved previously published genome sequencing reads from members of the Phyllopharyngea class [17,18] and generated *de novo* genome assemblies for seven species–*Chilodochona* sp., *Chilodonella uncinata*, *Chilodontopsis depressa*, *Dysteria derouxi*, *Hartmannula sinica*, *Trithigmostoma cucullulus*, and *Trochilia petrani* (S1 Table). We cleaned up each assembly to remove sequences from contaminants and predicted their genetic codes using the same methods as above. *Chilodochona* sp., *Chilodonella uncinata*, *Chilodontopsis depressa*, *Dysteria derouxi*, and *Trithigmostoma cucullulus* were all predicted to use the standard genetic code with both methods (S1 Fig). Surprisingly, however, we predicted that *H. sinica* and *T. petrani* use a variant genetic code, where the UAG stop codon has been reassigned to code for glutamine. Codetta identified 1489 and 350 UAG codons with a Pfam alignment (S2 Table) in *H. sinica* and *T. petrani*, respectively, and predicted that they translate UAG to glutamine with a log decoding probability of zero (S1 Fig). The number of UAG codons with an alignment was within the range of other sense codons (S2 Table). This

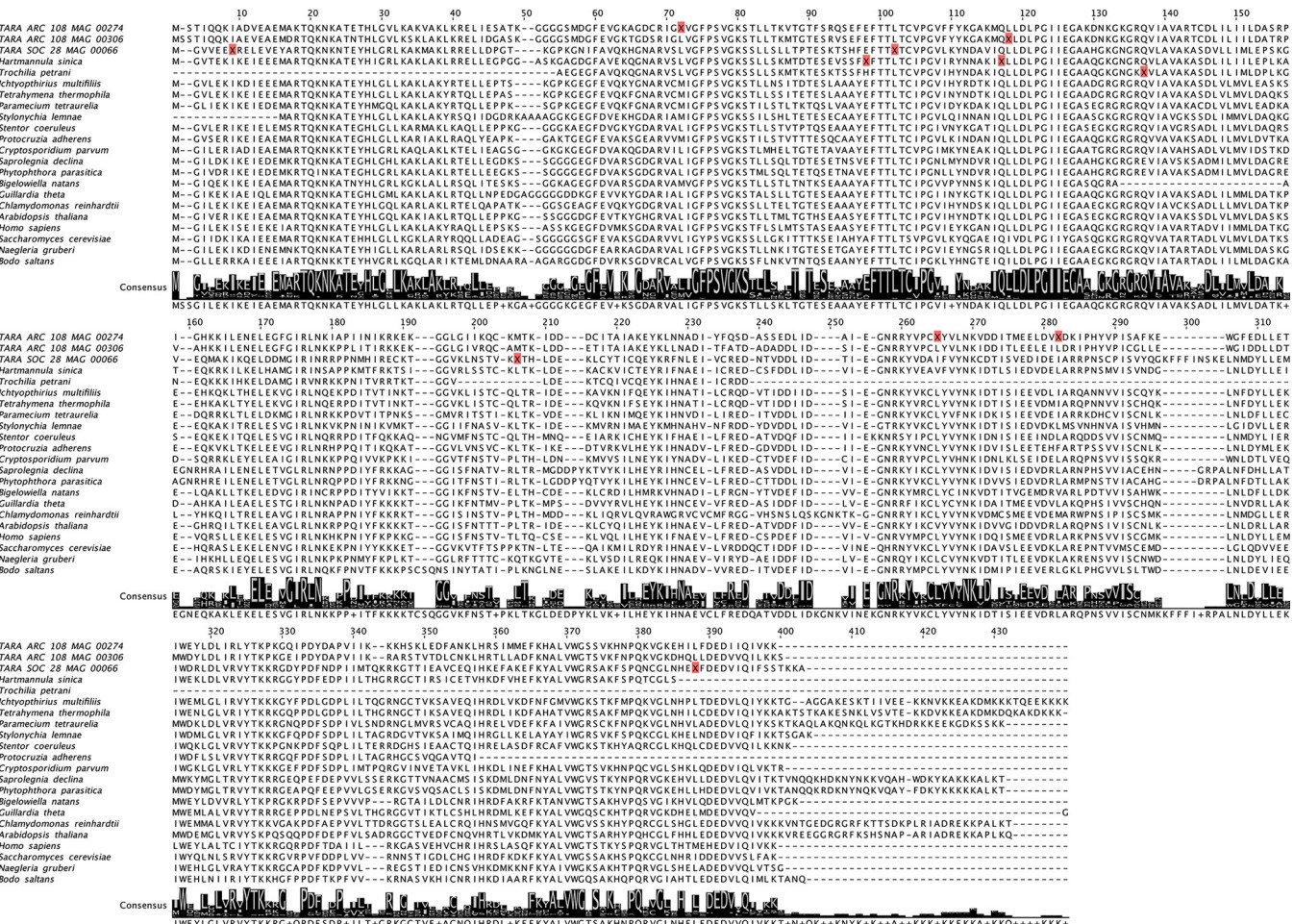

**Fig 2. Example multiple sequence alignment of orthologs of the DRG2 protein (developmentally-regulated GTP-binding protein 2) from ciliates and diverse representatives across Eukaryota.** Internal UAG codons are indicated by "X" with a red background. The in-frame UAG codons in the TARA MAGs occur at positions where leucine is highly conserved, whereas the in-frame UAG codons in *Hartmannula sinica* and *Trochilia petrani* occur at positions where glutamine is highly conserved.

prediction was supported by PhyloFisher which identified 30 and 7 internal UAG codons in *H. sinica* and *T. petrani*, respectively, which correspond to positions where glutamine is highly conserved (**S2 Fig**). Furthermore, our manual analysis of predicted gene models and ciliate orthogroups identified 363 and 130 internal UAG codons in *H. sinica* and *T. petrani*, respectively, of which 58.1% and 70.0% correspond to positions where glutamine is highly conserved (**Fig 1**). This genetic code variant (UAG translated to glutamine) is represented by translation table 15 and named the "Blepharisma Nuclear Code" in the NCBI database of genetic codes (https://www.ncbi.nlm.nih.gov/Taxonomy/Utils/wprintgc.cgi). We note that this name is incorrect and misleading given that *Blepharisma* do not use this genetic code variant.

As above, we investigated whether UAG has a dual context dependent meaning in *H. sinica* and *T. petrani* by aligning *Tetrahymena thermophila* reference proteins against their genome assemblies with miniprot. When translation table 1 was used, 93.5% - 100% of C-terminal protein alignments corresponded to the UAA stop codon (**S3 Table**). 0 alignments in *T. petrani* and only 5 in *H. sinica* corresponded to UAG codons, which was comparable to the number corresponding to other sense codons. When translation table 15 was used (i.e., treating UAA and UGA as stop codons and UAG as glutamine), the number of alignments where then end

of the query protein aligned increased. Furthermore, the distribution of UAG codons across the gene body showed a frequency pattern similar to that of synonymous glutamine codons (**S3 Fig**). Thus, it is unlikely that UAG has a dual context dependent meaning in *H. sinica* and *T. petrani*.

The partial *T. petrani* DRG2 sequence contains a single internal UAG codon corresponding to glutamine and the *H. sinica* DRG2 sequence contains a single internal UAG codon corresponding to glutamine (and one to glutamate) (**Fig 2**). The five species with the standard genetic code had high BUSCO completeness (72.5% to 82.5%) but the two species with reassigned UAG codons had low completeness (30.4% for *H. sinica* and 8.8% for *T. petrani*) (**S1 Table**).

## Phylogenomics reveals three independent genetic codon reassignments in Phyllopharyngea

To better understand the evolutionary relationships of the three uncultivated TARA MAGs, and to characterise the order of events surrounding the novel genetic code reassignments, we carried out a phylogenomics analysis focused on members of the CONthreeP lineage of ciliates–Colpodea, Oligohymenophorea, Nassophorea, Phyllopharyngea, Plagiopylea, and Prostomatea (which is not represented here). A concatenated alignment of 115 BUSCO proteins (53,648 amino acid sites after alignment trimming) from 29 ciliate species was constructed and used for phylogenomic analyses. Phylogenomic reconstruction was performed using maximum-likelihood (ML) and Bayesian approaches. The ML analysis was conducted using IQ-TREE under the LG+C20+F+G+PMSF model with 100 non-parametric bootstraps, while the Bayesian analysis was conducted using PhyloBayes MPI under the CAT-GTR model. Both methods yielded robust phylogenies with identical topologies and all branches had full statistical support from both methods (i.e., ML bootstrap support of 100% and a Bayesian posterior probability of 1) (**Fig 3**). Our phylogeny is in agreement with previous phylogenies based on small subunit ribosomal rRNA genes [18,21].

The three TARA MAGs formed a monophyletic lineage within Phyllopharyngea (**Fig 3**), confirming that they belong to the Phyllopharyngea class. This suggests that the reassignment of UAG to leucine occurred once in an ancestor of this lineage. ARC_306 grouped as sister to SOC_66, to the exclusion of ARC_274, despite geographical differences. *T. petrani* and *Dysteria derouxi* were grouped as sister lineages, as were *H. sinica* and *Chilodochona* sp. (**Fig 3**). This suggests that reassignment of UAG to glutamine independently evolved twice within the sampled Phyllopharyngea species and that these genetic code changes were more recent than the reassignment of UAG to leucine in the TARA MAGs lineage. The phylogenetic distribution of species that use the standard genetic code suggests that the most recent common ancestor of Phyllopharyngea used the standard genetic code (**Fig 3**).

## Novel suppressor tRNAs for UAG

Translation of the UAG codon to an amino acid requires a tRNA gene that can decode the UAG codon. We annotated tRNA genes in our dataset using tRNAscan-SE [22]. A single suppressor tRNA gene was identified in the ARC_274 MAG with a CUA anticodon which is predicted with high confidence to function as a leucine tRNA (**S5A Fig**). This tRNA gene is located on a 12.5 kb contig, capped at both ends with ciliate telomeric repeats (CCCCAAA/ GGGGTTT). Two suppressor tRNA genes were identified in the ARC_306 MAG with CUA anticodons that were predicted with high confidence to function as leucine tRNAs (**S5A Fig**). Both of these suppressor tRNA genes are located on the same 13.5 kb contig within 100 bp of each other but are not identical (75% identical) (**S5B Fig**). This contig has three gene models with best BLAST hits to ciliate sequences. One of the ARC_306 suppressor tRNAs is 94%

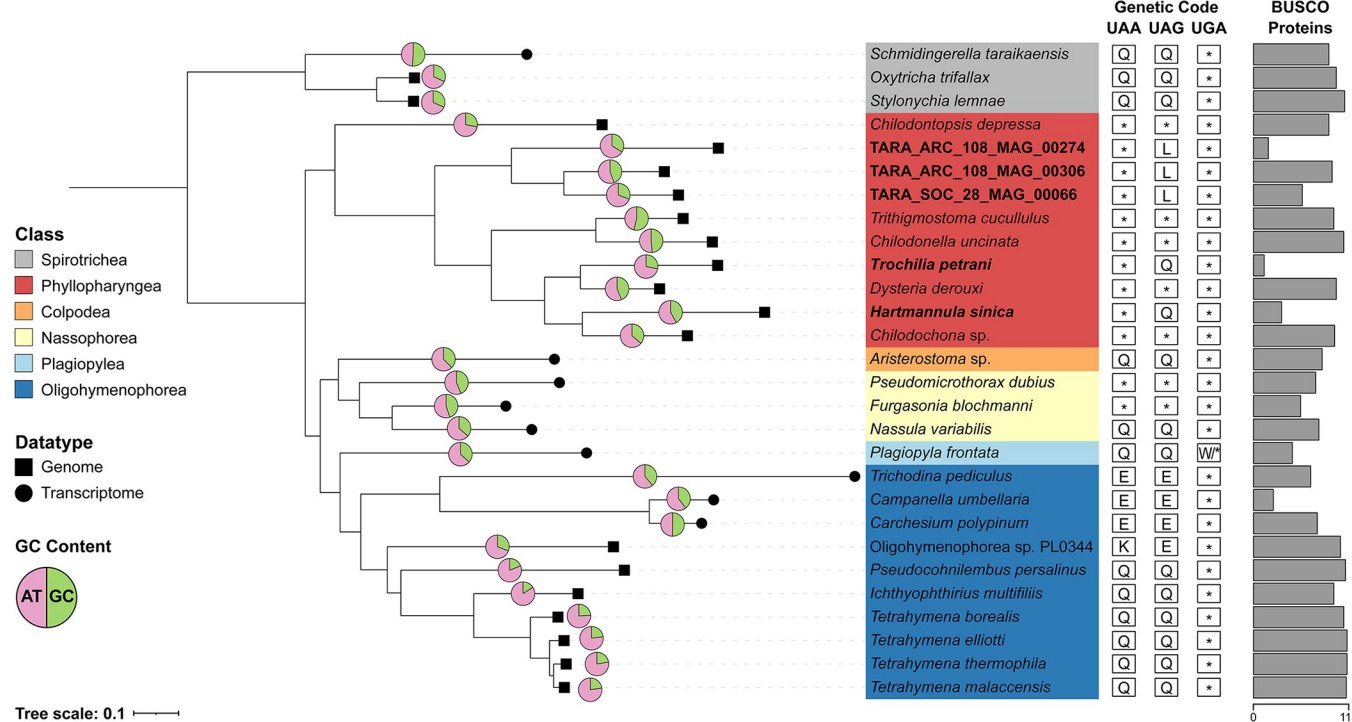

**Fig 3. Phylogenomics analysis of the CONthreeP lineage of ciliates.** The phylogeny was reconstructed from a concatenated alignment of 115 BUSCO proteins (53,648 amino acid sites). Maximum-likelihood analysis was conducted using IQ-Tree under the LG+C20+F+G model with PMSF approximation and 100 non-parametric bootstraps. Bayesian inference was performed using PhyloBayes MPI using the CAT-GTR model. The branch lengths displayed are from the ML analysis. All branches have full statistical support from both methods (i.e., ML bootstrap support of 100% and a Bayesian posterior probability of 1). *Oxytricha trifallax*, *Schmidingerella taraikaensis*, and *Stylonychia lemnae* are included as outgroups from the Spirotrichea class. The type of data (genomic or transcriptomic) is indicated by symbols at branch tips. GC content for each species is shown in a pie chart. Note that caution is required when comparing GC content between genome and transcriptome assemblies. The number of BUSCO proteins included in the concatenated alignment is shown in the bar pot, highlighting the amount of missing data per species. Genetic code changes are shown (*, STOP; Q, glutamine; L, leucine; K, lysine; W, tryptophan; E, glutamic acid).

identical to the suppressor tRNA from ARC_274, with only five nucleotide differences between the two sequences (one substitution in the anticodon loop and four substitutions in the variable loop) (**S5A and S5B Fig**). We compared the suppressor tRNA gene sequences against other phyllopharyngean tRNA sequences in our dataset which revealed that they are most similar in sequence to leucine tRNAs with CAA or TAA anticodons (**S5B Fig**). This suggests that the novel suppressor tRNAs evolved from a canonical leucine tRNA. Furthermore, the suppressor tRNA sequences contain many of the known identity elements for eukaryotic leucine tRNAs [23], including the discriminator base $A_{72}$, the positive identity determinants $A_4$-$U_{69}$ (not in the second ARC_306 tRNA-Sup which has $U_4$-$A_{69}$), $G_5$-$C_{68}$, a large variable region, and the invariant nucleotides $G_{18}$, $G_{19}$, $A_{21}$, $U_{33}$, $U_{54}$, $U_{55}$, $C_{56}$, and $A_{58}$ (**S5 Fig**), suggesting that they are likely to be recognised by leucyl-tRNA synthetases. We did not detect suppressor tRNA genes in the SOC_66 MAG, *H. sinica*, or *T. petrani*, however our analysis is likely limited by the low completeness of these assemblies. As expected, suppressor tRNA genes were not detected in the five genomes that use the standard genetic code.

## Codon usage of UAA, UAG, and UGA

To better understand the events preceding a genetic code change event, we annotated all 10 phyllopharyngean genomes and investigated usage of the standard stop codons and the

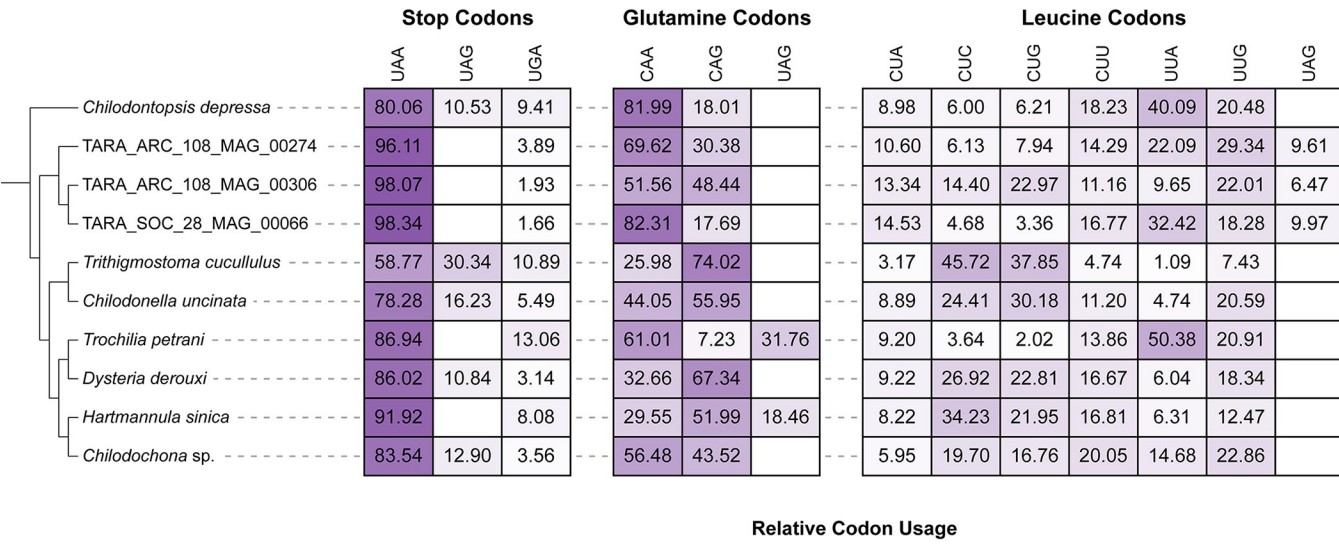

**Fig 4. Codon usage of UAA, UAG, and UGA codons and standard glutamine and leucine codons.** The heatmap shows the relative usage of each codon. The cladogram depicts the phylogenetic relationships determined in our phylogenomic analysis in **Fig 3**.

reassigned UAG codon following a genetic code change. We restricted this analysis to exclude partial gene models by only considering genes with both a predicted start and stop codon. UAA is the most used stop codon and UGA is the least used stop codon in all species in our dataset (**Fig 4**). *Trithigmostoma cucullulus* is a clear outlier in terms of codon usage with 58.8% of genes using UAA, 30.3% using UAG, and 10.9% using UGA as stop codons (**Fig 4**), which is reflected by the higher GC content of its genome (**Fig 3** and **S1 Table**). UAA is used as stop codon in 78.3%– 86% of genes in the other species that use the standard genetic code (**Fig 4**). In the species that have retained UAG as a stop codon (excluding *Trithigmostoma*), UAG is used as stop codons for 10.5%– 16.2% of genes (**Fig 4**). This suggests that UAG usage was low but non-negligible prior to the genetic code change events. UGA is used less frequently in these species, with only 3.1%– 9.4% of genes using UGA as a stop codon (**Fig 4**).

Usage of UAA as a stop codon is increased to 96.1%– 98.3% of genes in the TARA MAGs following their genetic code reassignment (**Fig 4**). Fewer than 3.9% of genes use UGA as a stop codon in these MAGs. UAA usage is also increased in *H. sinica* compared to its closest relative (91.9% vs 83.5%) (**Fig 4**). Likewise, UGA usage increased in *H. sinica* (8.1% vs 3.6%) and *T. petrani* compared to their closest relatives (13.1% vs 3.1%) following their genetic code reassignments (**Fig 4**).

In *T. petrani* and *H. sinica*, we compared relative codon usage of the reassigned UAG codon compared to the two standard glutamine codons (CAA and CAG). The reassigned UAG codon is the second most used glutamine codon in *T. petrani* (31.8%) but the least used in *H.sinica* (18.5%) (**Fig 4**). 83.7% and 78.2% of genes in *T. petrani* and *H. sinica* contain at least one internal in-frame UAG codon, showing that the reassigned codon is widely used in both species. In the three TARA MAGs, we compared usage of the reassigned UAG codon compared to the six standard leucine codons (CUA, CUC, CUG, CUU, UUA, and UUG). Relative codon usage of the reassigned UAG codon ranges from 6.5% to 10% in the TARA MAGs (**Fig 4**). 81.3% to 91.1% of genes contain at least one internal UAG codon in these MAGs, showing that usage of the reassigned codon is also widespread in these genomes.

## Discussion

Here we report the identification of three novel genetic code changes in ciliates. We discovered that three uncultivated ciliates sequenced from eukaryotic metagenomes by the TARA Oceans Project [16] use a variant genetic code where the UAG stop codon has been reassigned to encode leucine (**Fig 1**). Phylogenomic analysis revealed that the uncultivated ciliates belong to the Phyllopharyngea class. Reassembly and analysis of seven other phyllopharyngean genomes led to the discovery of another two genetic code changes in this lineage–reassignment of UAG to glutamine in both *H. sinica* and *T. petrani* (**Fig 1**). In contrast to recent reports that dual context dependent or "bifunctional" stop codons are prevalent in ciliates[7], we found no evidence to suggest that UAG has a dual context dependent meaning in the Phyllopharyngea class of ciliates.

It is important to note that the genetic code changes reported herein, and indeed most published reports of genetic code changes, are predictions based on genomic data. While these are high-confidence predictions with multiple lines of evidence including large numbers of internal in-frame UAG codons that occur at conserved leucine/glutamine positions and the identification of predicted cognate suppressor tRNA genes in two genome assemblies, confirmation that a codon is translated to a particular amino acid requires proteome sequencing (e.g., mass spectrometry) and comparison with corresponding coding sequences.

Given the complex phylogenetic distribution of the standard and variant genetic codes in the ciliate lineage [10], there are conflicting interpretations surrounding the order of events. Either (1) independent genetic code changes occurred in multiple ciliate lineages, including different lineages convergently evolving the same genetic code variant, or (2) stop codon reassignments evolved in more ancient lineages of ciliates giving rise to the taxa with variant genetic codes, which were followed by reversions to the original function as stop codons in the taxa that use the standard genetic code. A recent study proposed that ancestral ciliates reassigned all three stop codons as sense codons, followed by one or more of the reassigned codons reverting to functioning as stop codons giving rise to the different types of genetic codes observed in ciliates (i.e., the standard genetic code or genetic codes with one or more reassigned stop codons) [7]. In our study, we focus just on the Phyllopharyngea class of ciliates. From these phylogenomic analyses, the most parsimonious explanation for the distribution of genetic codes within sampled phyllopharyngean ciliates is that the most recent common ancestor of Phyllopharyngea used the standard genetic code (**Fig 3**). This lineage then underwent at least three independent genetic code change events based on sampled species: (1) reassignment of UAG to leucine in the lineage of uncultivated TARA MAGs, (2) reassignment of UAG to glutamine in the lineage giving rise to *H. sinica*, and (3) reassignment of UAG to glutamine in the lineage giving rise to *T. petrani* (**Fig 3**). Given the widespread usage of the reassigned UAG codons in Phyllopharyngea, with 78.2% to 91.1% of genes using a least one UAG codon, it is unlikely that UAG will revert to functioning as a stop codon. If it were to revert, it would result in large-scale protein truncation due to the presence of internal in-frame UAG codons in most genes unless there were genome-wide substitutions of UAG to another sense codon (e.g., due to genomic GC content shifting). Thus, we propose that it is unlikely that a reversion could so readily occur once a stop codon has been reassigned to a sense codon.

Several hypotheses have been proposed to model the processes surrounding genetic code changes. Under the "codon capture" hypothesis, a codon is driven to extinction by mutational biases (e.g., low GC-content) followed by loss of the corresponding tRNA (or loss of function in release factors in the case of stop codons) [24]. This unused codon could later be captured by a noncognate tRNA and reappear in the genome, resulting in a change to the genetic code. The "ambiguous intermediate" hypothesis proposes that genetic code changes involve an

intermediate stage where a codon is ambiguously translated via competing tRNAs charged with different amino acids [25]. In the scenario of a stop codon being reassigned, this would involve a suppressor tRNA competing with a release factor protein. The "unassigned codon" hypothesis [26] and the "tRNA loss driven codon reassignment" [27] hypothesis propose that the genetic code change is preceded by loss of function or reduced efficiency of a tRNA or release factor, resulting in an unassigned or inefficient codon that can be captured by another tRNA gene.

It is still unclear what has driven ciliates to evolve so many genetic code variants and why there is differential retention of UAA/UAG/UGA as stop codons. Our analysis of codon usage shows that usage of UAG as a stop codon is low but non-negligible (mean 16.2% of genes) in phyllopharyngean species that use the standard genetic code (**Fig 4**). UGA is the least used stop codon in all taxa in our dataset (1.7% - 13.1% of genes) (**Fig 4**). Thus, it is unclear why UGA was retained as a stop codon but not UAG. Furthermore, the UGA codon is known to be the least robust and most prone to translational readthrough [28] confounding the situation further. Likewise, the low GC content that is a common characteristic of extant ciliate genomes does not explain their propensity to evolve non-standard genetic codes. GC content varies considerably within Phyllopharyngea (27.4% to 53.6%) (**Fig 3**). *Chilodontopsis depressa* has a lower GC content (28%) than most of the other species in our dataset but uses the standard genetic code and has similar stop codon usage as other species with higher GC content (**Figs 3 and 4**).

The evolution of the UAA and UAG codons are thought to be coupled as they virtually always have the same meaning–either they both function as stop codons or they are both reassigned to code for the same amino acid [11]. The first reported cases where UAA and UAG have different meanings in a nuclear genome were reported in two unrelated taxa–an uncultured rhizarian where UAG was reassigned to code for leucine and in the fornicate *Iotanema spirale* where UAG was recoded for glutamine [12]. Another variant was reported in green algal genus *Scotinosphaera* where UAG encodes serine but UAA is retained as a stop codon [29]. Recently, we reported a novel genetic code variant in an uncultured ciliate, where the UAA and UAG codons were reassigned to code for two different amino acids (lysine and glutamic acid, respectively), the first reported case where UAA and UAG encode two different amino acids [10]. In this study, we report another deviation from the trend of UAA and UAG having the same function. We show that UAG has been reassigned to function as a sense codon, but UAA was retained as a stop codon in five of the studied genomes. Furthermore, we show that it is unlikely that the UAA stop codon could later be reassigned to an amino acid in these lineages, as seen in most other genetic code variants, given the almost ubiquitous usage of UAA as the preferred stop codon with 86.9% to 98.3% of genes using UAA as a stop codon (**Fig 4**). If UAA were to be later reassigned to function as a sense codon, it would result in widespread protein elongation with proteins extending downstream to the next in-frame stop codon (i.e., UGA). Ciliates typically have very short 3'-UTRs which would somewhat limit the effect, but this would still impact almost the entire proteome. Such levels of protein elongation would likely have similar deleterious consequences as widespread translational readthrough, including issues with protein aggregation and stability, disruption to localisation signals and energetic waste [30]. Thus, the function of UAA as the preferred stop codon is likely fixed in these taxa. Had the UAA codon been reassigned initially, it likely would have also triggered the UAG codon to be reassigned to encode the same amino acid given that a suppressor tRNA that decodes UAA is also expected to decode UAG due to wobble base pairing of the UUA anticodon [13]. This is not the case when UAG is reassigned on its own, as a suppressor tRNA with an anticodon complementary to UAG (i.e., CUA anticodon) is not expected to recognise the UAA codon, allowing the UAG codon to evolve independently of UAA.

Our results highlight the evolvability of the genetic code and the tendency, not only for ciliates but also unrelated taxa, to independently evolve the same genetic code variants. Multiple lineages within Ciliophora have independently evolved the translation of UAR codons to glutamine [10,31]. Translation of UAR to glutamine is also found in the nuclear genomes of green algae from the Ulvophyceae class [32], the diplomonad *Hexamita* [33], the oxymonad *Streblomastix strix* [34], and the aphelid *Amoeboaphelidium protococcarum* [35]. Similarly, reassignment of the UAG stop codon (but not UAA) to leucine is reported here for the first time in ciliates but was previously reported in the nuclear genome of an uncultured rhizarian [12] and also in the mitochondrial genomes of chlorophyte algae [36,37] and in some chytridiomycete fungi [38]. Reassignment of UAG (but not UAA) to glutamine is also reported here to have evolved independently twice within sampled phyllopharyngean ciliates and was also previously reported in the distantly related ciliate *Protocruzia tuzeti* [7] and the fornicate *Iotanema spirale* [12]. These findings highlight that while the genetic code is one of the most conserved features of molecular biology, it is not quite as universal as was once thought [39]. Genetic code reassignments are relatively recent events, demonstrating that genetic code evolution is an ongoing process.

## Materials and methods

### Dataset assembly

Ciliate MAGs from the TARA Oceans project were downloaded from https://www.genoscope.cns.fr/tara/ [16]. The three MAGs focused on in this study are TARA_ARC_108_MAG_00274, TARA_ARC_108_MAG_00306, and TARA_SOC_28_MAG_00066. Genome sequencing reads for *Chilodochona* sp. (SRR9841583), *Chilodontopsis depressa* (SRR9841577), *Dysteria derouxi* (SRR9841578), *Hartmannula sinica* (SRR9841582), *Trithigmostoma cucullulus* (SRR9841579), and *Trochilia petrani* (SRR9841580) were downloaded from BioProject PRJNA546036 [18]. Genome sequencing reads for *Chilodonella uncinata* (SRR6195035) were downloaded from BioProject PRJNA413041 [17].

Genome assemblies were generated using SPAdes (v3.15.5) [40] with default settings, except single-cell (—sc) mode was enabled. Contigs shorter than 1000 bp were discarded. Assemblies were decontaminated using a combination of Tiara [41] and contig clustering based on tetranucleotide frequencies.

### Genetic code prediction and genome annotation

The genetic code used by each genome was initially predicted using Codetta (v2.0) [20] and the PhyloFisher "genetic_code_examiner" utility [19]. For the five phyllopharyngean species that use the standard genetic code, an initial gene set was generated using GeneMark-EP [42], with hints generated by ProtHint from a database of 1,170,806 Alveolata proteins. These initial gene sets were filtered by selecting complete gene models (i.e., containing a start and stop codon) that had full-length alignments (alignment length ≥ 95% of both the query and subject sequence lengths) against the Alveolata protein database using Diamond in ultra-sensitive mode [43]. These filtered subsets were used as training gene sets to train an Augustus model for each species [44]. This process was repeated by incorporating the initial Augustus gene models from every other species into the protein database supplied to GeneMark-EP and to retrain Augustus and generate a final gene set for each species. For the five species that use variant genetic codes, an initial gene set was generated using the most appropriate Augustus model from above (with modified parameters such that UAG is no longer used as a stop codon), which went through a similar filtering step to select training genes to train a species-specific Augustus model for each genome.

The final gene sets were used to further interrogate the genetic code. In-frame UAG codons were translated to "X". Orthogroups from a dataset of 19 ciliate species (including the 10 phyllopharyngean genomes) (**S1 Table**) were identified using OrthoFinder (v.2.5.5) [45] with the parameter "-M msa". A multiple sequence alignment was generated for each orthogroup using MAFFT (v7.520) [46]. In-frame UAG codons that correspond to highly conserved amino acid positions ($\geq$ 70% identity) in aligned orthogroups were identified and the most numerous amino acid at these sites were counted. The counts were visualised as a sequence logo using WebLogo (v3.7.12) [47]. This genetic code analysis and subsequent analyses of codon usage were restricted to gene models with predicted start and stop codons (i.e., not partial gene models). tRNA genes were annotated using tRNAscan-SE (v.2.0.7) in eukaryotic search mode [22].

To investigate if UAG has a dual context dependent meaning in phyllopharyngean ciliates using a method independent of our genome annotation, we aligned reference *Tetrahymena thermophila* proteins against each genome assembly using miniprot (v0.13) [48] with different translation tables (1/6/15/16). Alignments were filtered to identify cases where the end of query proteins aligned to the genome and the codons at these corresponding positions were counted (**S3 Table**).

## Phylogenetics of alpha-tubulin sequences

An alpha-tubulin gene was recovered from ARC_306 and used for phylogenetic analysis, using a dataset of alpha-tubulin sequences from a previously published phylogenetics study of ciliates [49]. Three apicomplexan sequences were included as outgroups. Sequences were aligned using MAFFT with the L-INS-I algorithm [46]. A maximum-likelihood phylogeny was constructed using IQ-Tree (v2.2.2.6) [50] under the LG+R3 model which was the best fitting model according to ModelFinder [51]. Support was assessed using 1000 ultrafast bootstrap replicates [52].

## Phylogenomics

We constructed a dataset of ciliate genomes and transcriptomes for phylogenomic analyses, focusing on members of the CONthreeP lineage (**Fig 3**). *Oxytricha trifallax*, *Schmidingerella taraikaensis*, and *Stylonychia lemnae* from the Spirotrichea class were included as outgroups. A concatenated supermatrix of BUSCO proteins was generated using our BUSCO_phylogenomics pipeline (https://github.com/jamiemcg/BUSCO_phylogenomics). 115 BUSCO proteins from the Alveolata_odb10 dataset [53] were identified as complete and single copy in at least 60% of species and were included in our analysis. Each BUSCO family was individually aligned using MUSCLE (v5.1) [54]. Alignments were then trimmed using trimAl (v1.4) [55] with the "automated1" parameter and concatenated together resulting in a supermatrix alignment of 53,648 sites. Maximum-likelihood phylogenomic reconstruction was performed using IQ-TREE (v2.2.2.6) [50]. Most sequences (26/29) failed IQ-TREE's chi-square test for homogeneity of character composition, therefore we carried out maximum-likelihood analysis under the C20 protein mixture model (LG+C20+F+G model) with PMSF approximation [56] and 100 non-parametric bootstraps, using a guide tree from FastTree (v2.1.11) [57]. Bayesian analyses were also conducted using PhyloBayes MPI (v1.8) [58] under the CAT-GTR model. Two independent Markov chain Monte Carlo chains were run for approximately 10,000 generations. Convergence was assessed using bpcomp and tracecomp, with a burn-in of 20%. Phylogenies were visualised and annotated using iTOL [59].

## Supporting information

**S1 Fig. Genetic code prediction of the UAG codon using Codetta.** The table shows log decoding probabilities of UAG for each amino acid. "?" indicates that there were insufficient

alignments to infer an amino acid meaning (which is the expected behaviour for stop codons).
(PDF)

**S2 Fig. Genetic code prediction of the UAG codon using the genetic_code_examiner utility from PhyloFisher.**
(PDF)

**S3 Fig. Frequency of in-frame UAG codons across gene body compared to standard leucine and glutamine codons.**
(PDF)

**S4 Fig. Maximum-likelihood phylogeny of 104 ciliate alpha-tubulin sequences constructed using IQ-TREE under the LG+R3 model.** Numbers represent support from 1000 ultrafast bootstrap replicates. Three apicomplexan sequences were included as an outgroup.
(PDF)

**S5 Fig. (A)** Three suppressor tRNA genes from two TARA Oceans ciliate MAGs (ARC_274 and ARC_306) with CUA anticodons that are predicted to function as leucine tRNAs. **(B)** Multiple sequence alignment of the three suppressor tRNA genes with representative canonical leucine tRNAs with CAA and TAA anticodons.
(PDF)

**S1 Table. Summary of the genome assembly and annotation statistics and the species included in the OrthoFinder analysis.**
(XLSX)

**S2 Table. Number of Pfam alignments per codon from the Codetta analysis.**
(XLSX)

**S3 Table. Counts of miniprot alignments of reference proteins against each genome assembly corresponding to stop codons.**
(XLSX)

## Acknowledgments

The authors acknowledge the work delivered via the Research Computing Group at EI who manage and deliver High Performance Computing at EI.

## Author Contributions

**Conceptualization:** Jamie McGowan, Thomas A. Richards, Neil Hall, David Swarbreck.

**Data curation:** Jamie McGowan.

**Formal analysis:** Jamie McGowan.

**Funding acquisition:** Thomas A. Richards, Neil Hall, David Swarbreck.

**Investigation:** Jamie McGowan.

**Methodology:** Jamie McGowan.

**Software:** Jamie McGowan.

**Supervision:** David Swarbreck.

**Validation:** Jamie McGowan.

**Visualization:** Jamie McGowan.

**Writing – original draft:** Jamie McGowan, Thomas A. Richards, Neil Hall, David Swarbreck.

**Writing – review & editing:** Jamie McGowan, Thomas A. Richards, Neil Hall, David Swarbreck.

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
