## [Decision Letter · Decision Letter 0]

4 Nov 2024

PGENETICS-D-24-00945Multiple Independent Genetic Code Reassignments of the UAG Stop Codon in Phyllopharyngean CiliatesPLOS Genetics Dear Dr. McGowan, Thank you for submitting your manuscript to PLOS Genetics. After careful consideration, we feel that it has merit but does not fully meet PLOS Genetics's publication criteria as it currently stands. Therefore, we invite you to submit a revised version of the manuscript that addresses the points raised during the review process. Please submit your revised manuscript within 30 days Dec 04 2024 11:59PM. If you will need more time than this to complete your revisions, please reply to this message or contact the journal office at plosgenetics@plos.org. Please include the following items when submitting your revised manuscript:*
A rebuttal letter that responds to each point raised by the editor and reviewer(s). You should upload this letter as a separate file labeled 'Response to Reviewers'. This file does not need to include responses to formatting updates and technical items listed in the 'Journal Requirements' section below.*
A marked-up copy of your manuscript that highlights changes made to the original version. You should upload this as a separate file labeled 'Revised Manuscript with Track Changes'.*
An unmarked version of your revised paper without tracked changes. You should upload this as a separate file labeled 'Manuscript'. If you would like to make changes to your financial disclosure, competing interests statement, or data availability statement, please make these updates within the submission form at the time of resubmission. Guidelines for resubmitting your figure files are available below the reviewer comments at the end of this letter. We look forward to receiving your revised manuscript. Kind regards, Laurent DuretGuest EditorPLOS Genetics Eva StukenbrockSection EditorPLOS Genetics Aimée DudleyEditor-in-ChiefPLOS Genetics Anne GorielyEditor-in-ChiefPLOS Genetics **Journal Requirements:** **Additional Editor Comments (if provided):** Dear Jamie McGowan,

Thank you for your patience while your manuscript "Multiple Independent Genetic Code Reassignments of the UAG Stop Codon in Phyllopharyngean Ciliates" was peer-reviewed at PLOS Genetics. It has now been evaluated by three independent reviewers.

As you will see, the three reviewers are very positive and consider that your findings are an important contribution to the field of genetic code evolution.

They made several comments that will be helpful to improve your manuscript. In particular, as pointed out by Reviewer #1, it would be important to clarify whether UAG is completely reassigned or whether it may still be used as a stop codon to some degree.

In light of the reviews, we would like to invite you to revise the work to carefully address the reviewers' reports.

Laurent Duret**Reviewers' comments:** Reviewer's Responses to Questions

**Comments to the Authors:**

Reviewer #1: In this paper, McGowan et al report the discovery of novel reassignments of the UAG stop codon in the ciliate class Phyllopharyngea. UAG appears to be reassigned to leucine in a clade of uncultivated ciliates only known from metagenomics, and UAG is additionally reassigned to glutamine in two other Phyllopharyngeal lineages represented by a single species each. These findings add to the already extensive diversity of stop codon reassignments in ciliates, a fascinating system with regards to genetic code evolution. Neither reassignment of UAG to leucine or glutamine in isolation has been previously reported in ciliates. Broadening the diversity of stop codon reassignments known in ciliates is important for future work on why stop codons reassignments are so prevalent in the ciliate clade and on the features that determine codon specificity of eukaryotic release factors.

This claims in this paper are supported by rigorous computational methodology. Since these species are uncultivated, experimental confirmation of these reassignments is out of reach and having multiple lines of computational support is critical. Genetic code predictions were validated by three independent computational approaches using different sets of conserved genes. A consistent suppressor tRNA is found in two of the genomes. Phylogenetic analyses repeated using both ML and Bayesian approaches to establish that the UAG to glutamine reassignments are truly two independent events.

Major comment

- The authors should clarify whether UAG is completely reassigned or whether it may still be used as a stop codon to some degree. These reassignments are currently referred to as if they are unambiguous stop-to-sense reassignments, but no evidence is shown for whether UAG is or isn’t still used as a stop. This is an important point to address in ciliates where dual-sense stop codons have been found in at least two independent lineages. This could be established by an analysis looking at whether UAGs ever occur at positions that align to termination codons in related ciliates. Another analysis that may be interesting to see for the reassigned species is whether UAG usage as a sense codon is depleted in the proximity of the real termination codon (see Fig 5A of Seah et al, 2022 https://doi.org/10.24072/pcjournal.141 ), which would indicate some remaining context-dependent termination at UAGs. Knowing whether UAG still functions as a stop codon is important for understanding how this reassignment evolved and what mutations to the translational machinery would have been necessary to allow for it.

Minor comments:

- For the computational genetic code predictions by the various methods, it may be informative to mention the number of codons for the average sense codon and for the other stop codons UAA and UGA, just to establish a reference point for whether the number of in-frame UAGs is similar to other sense codons. Alignment-based approaches could potentially get spurious alignment of stop codons to protein models from recent pseudogenes or sequencing errors.

- In figure 4, dual-sense stop codons are not indicated, making it seem like all of these stop-to-sense reassignments are unambiguous. For example, Plagiopyla frontata, as indicated your previous paper (McGowan et al, 2023), has UGA -> W/*

- In the section about codon usage (lines 225-256), it is unclear how stop codon usage was calculated. For species with predicted UAG reassignments, was UAG considered to be a potential stop codon?

- In lines 216-218, the statement that the suppressor tRNAs are most similar to leucine tRNAs needs to be substantiated more. How is this similarity measured? What other tRNAs were the suppressor tRNAs compared against? How were the tRNA alignments constructed? You can also refer to the literature on tRNA identity to argue that these tRNAs are likely recognized by the leucyl-tRNA synthetase due to the presence of tRNA features (see Giege et al, 2023 https://doi.org/10.1093/nar/gkad007 ).

- In Figure S4a, the tRNA secondary structure for the TARA ARC 108 MAG 00306 SupCTA tRNA is incorrect. The variable stem-loop has no loop, which is physically impossible. Instead, the bulged “AA” should be base pairing with the UU, and there should be a GCAA tetraloop at the end. Similarly, the alignment in S4b should be adjusted (especially if arguments about sequence similarity are being based on it). In particular, in the D-loop, the invariant nucleotides G18 and G19 should be aligned to the same column for all sequences. There are designated positions where insertions in tRNAs should be placed (see Sprinzl et al, 1998 https://doi.org/10.1093/nar/26.1.148 ). A decent alignment can probably be automatically generated by tRNAscan-SE or Infernal. Currently there is no description of how this alignment was generated.

- In the discussion (lines 292-298), the argument that reverse reassignment back to a stop codon is unlikely is weak. All codon reassignments are deeply unlikely—part of what makes them so interesting is the fact that they do evolve despite being counterintuitive. One could imagine UAG reverting back to being a sense codon, for example, if genomic GC content shifted such that UAG became a very rare sense codon, and now UAG can be reassigned to a stop codon more easily.

Reviewer #2: I am attaching the review as MS Word file to preserve formatting of the text.

Reviewer #3: In this manuscript McGowan et al have analysed 30 metagenomic assemblies from TARA Oceans project and identified three genomes with UAG codon reassignment decoupled from UAA codon. [They have reported such decoupling in another species two years ago in a publication in the same journal.] Then they assembled additional genomes using raw data obtained from the related species and found additional reassignments. In total three genomes with UAG reassignment to L and two with reassignment to Q. The authors employed two methods for identification of codon reassignments that provide strong support for these findings.

By obtaining sequences from closely related species they were able to reconstruct their phylogenetic relationship and explore the evolution of these reassignments. The analysis suggests that UAG reassignments to Q have occurred independently at least twice. This specific claim relies on the limited genomic data and the accuracy of phylogenetic analysis. I find support for this specific claim relatively weak, but I agree with the authors’ interpretation as the most parsimonious scenario in the light of available data.

The manuscript is sufficiently detailed to enable reproduction of this study and is well written and easy to understand, albeit I have some minor suggestions for improving the text below.

In conclusion, I think it is an important and reliable study that expands our understanding of variant genetic codes diversity in ciliates and its evolution. My comments and suggestions below are intended mostly just to clarify some aspects of this study.

Specific comments:

1. The authors mentioned in Results that they cleaned the data from contaminant sequences before assembly. I couldn’t find the description of how this was done.

2. The authors report the number of UAG codons corresponding to inferred amino acids in pfam alignments. I still haven’t used Codetta myself and don’t know if it is easy/possible to get information on the number of pfam matches where UAG corresponds to a different amino acid. It would be interesting (of possible) to have Codetta analysis represented in the same way as PhyloFisher (Figure S2).

3. I would appreciate if the authors could provide a bit more detail on selection of specific models for phylogenetic analysis.

4. The authors discuss the reasons for UAG and UAA coupled meaning in variant genetic codes in the Discussion section and also argue that reassignment of UAA is less likely that of UAG, indeed it has not been observed so far. I fully agree with the authors argument, but I wonder if it will be useful to bring this to introduction to explain why UAG/UAA decoupling is so rare and interesting.

5. the use of canonical/non-canonical:

“In most reported non-canonical genetic code changes…” such wording may suggest that there are canonical and non-canonical changes of the genetic code. But I am sure that this is not what the authors meant. Overall, if possible, I would advice against using ‘canonical’ and ‘non-canonical’. Besides the theological origin of such language these are vague descriptors. Readers opinions on what is canonical and non-canonical may differ and also change over time. I would suggest using more concrete terms, i.e. ‘standard genetic code’ and ‘variant genetic codes’, the authors do use these terms at places, so it may be also better to stick to them for consistency. I find “Codon usage of canonical and reassigned stop codons” also a bit awkward, one may argue that UAG, UAA and UGA are standard (or canonical if you wish) stop codons and other stop codons (such as AGG in vertebrate mitochondria) are non-standard (non-canonical). But that’s npt what is meant here. I wonder if it will be more accurate to formulate it as something like this “Codon usage of UAA, UAG and UGA specifying stops or reassigned amino acids” or even simpler “UAA, UAG and UGA codons usage”.

6. Line 23

“an unusual exception” – to avoid accidental misinterpretation by a uninitiated reader of this being a unique exception, I would consider using a word other than “unusual”, say, something that would emphasize ciliates as being an extreme example of such exceptions for example “the most prominent exception”.

7. Line 196: Change “that translation of UAG to glutamine” to “that reassignment of UAG to glutamine”

Pasha Baranov

**Have all data underlying the figures and results presented in the manuscript been provided?**

Reviewer #1: Yes

Reviewer #2: Yes

Reviewer #3: Yes

PLOS authors have the option to publish the peer review history of their article (what does this mean?). If published, this will include your full peer review and any attached files.

Reviewer #1: No

Reviewer #2: No

Reviewer #3: **Yes: **Pavel Baranov

 **Figure resubmission:** While revising your submission, please upload your figure files to the Preflight Analysis and Conversion Engine (PACE) digital diagnostic tool, https://pacev2.apexcovantage.com/. PACE helps ensure that figures meet PLOS requirements. To use PACE, you must first register as a user. Registration is free. Then, login and navigate to the UPLOAD tab, where you will find detailed instructions on how to use the tool. If you encounter any issues or have any questions when using PACE, please email PLOS at figures@plos.org. Please note that Supporting Information files do not need this step. If there are other versions of figure files still present in your submission file inventory at resubmission, please replace them with the PACE-processed versions. **Reproducibility:** To enhance the reproducibility of your results, we recommend that authors deposit laboratory protocols in protocols.io, where a protocol can be assigned its own identifier (DOI) such that it can be cited independently in the future. Additionally, PLOS ONE offers an option to publish peer-reviewed clinical study protocols. Read more information on sharing protocols at https://plos.org/protocols?utm_medium=editorial-email&utm_source=authorletters&utm_campaign=protocols

---

## [Editor Report · Decision Letter 1]

25 Nov 2024

Dear Dr McGowan,

We are pleased to inform you that your manuscript entitled "Multiple Independent Genetic Code Reassignments of the UAG Stop Codon in Phyllopharyngean Ciliates" has been editorially accepted for publication in PLOS Genetics. Congratulations!

Yours sincerely,

Laurent Duret

Guest Editor

PLOS Genetics

Eva Stukenbrock

Section Editor

PLOS Genetics

Aimée Dudley

Editor-in-Chief

PLOS Genetics

Anne Goriely

Editor-in-Chief

PLOS Genetics

Comments from the reviewers (if applicable):

**Data Deposition**

http://datadryad.org/submit?journalID=pgenetics&manu=PGENETICS-D-24-00945R1

**Press Queries**

---

## [Editor Report · Acceptance letter]

10 Dec 2024

PGENETICS-D-24-00945R1 

Multiple Independent Genetic Code Reassignments of the UAG Stop Codon in Phyllopharyngean Ciliates 

Dear Dr McGowan, 

We are pleased to inform you that your manuscript entitled "Multiple Independent Genetic Code Reassignments of the UAG Stop Codon in Phyllopharyngean Ciliates" has been formally accepted for publication in PLOS Genetics! Your manuscript is now with our production department and you will be notified of the publication date in due course.

With kind regards,

Zsofia Freund

PLOS Genetics

On behalf of:
